# Characteristics of Cyclist Crashes Using Polytomous Latent Class Analysis and Bias-Reduced Logistic Regression

**Yuta Sekiguchi [1]**, **Masayoshi Tanishita [2,***]** **and Daisuke Sunaga [2]**

1   Graduate School of Science and Engineering, Chuo University, Tokyo 112-8551, Japan;
    a17.sr4m@g.chuo-u.ac.jp
2   Department of Civil and Environmental Engineering, Chuo University, Tokyo 112-8551, Japan;
    dsunaga.385@g.chuo-u.ac.jp
*   Correspondence: mtanishita.45e@g.chuo-u.ac.jp; Tel.: +81-3-3817-1810

**Abstract:** Although the number of cyclist crashes is decreasing in Japan, the fatality rate is not. Thus, reducing their severity is a major challenge. We used a polytomous latent class analysis to understand their characteristics and bias-reduced logistic regression to analyze their severity. Specifically, 90,696 combinations and 139,955 cyclist accidents were divided into 17 classes. The variable contributing the most to the classification was the crash location. Common fatality risks included older age groups and rural areas, whereas other factors differed among crash locations. Median strips, stop signs, and boundaries between the sidewalk and roadway affected the severity of crashes at intersections. Moreover, the existence of a median strip, collision partner, and time period affected the severity of crashes between intersections. On the sidewalks, the fatality risk was higher when the front part of the bicycle was subjected to the collision.

**Keywords:** cyclist crash; polytomous latent class analysis; bias-reduced logistic regression

## 1. Introduction

Cycling has various advantages, including the potential for diminishing obesity; promoting health; and reducing the noise, air pollution, and $CO_2$ emissions associated with travel. However, cyclists have a higher risk of death than motor vehicle occupants. Therefore, understanding how to make cycling safer amid increasing rates of cycling is important to improving people's well-being. There is a growing body of research examining the safety of cyclists and severity of cyclist crashes [1–17]. Recently, Macioszek and Granà [18] analyzed the severity of cyclist injuries in cyclist–vehicle crashes. Using a binary logit model, they showed that the factors affecting the crash severity were the vehicle drivers' genders and ages, driving under the influence of alcohol, vehicles exceeding the speed limit, cyclist ages, cycling under the influence of alcohol, the speed of the cyclist before the incident, vehicle type (truck), incident place (road), time of day, and incident type.

The characteristics with the potential to influence the severity studied by various authors (including the characteristics noted above) can be grouped into those concerning the bicycle, vehicle and vehicle driver, environment, and road infrastructure (including regulations and signs). Among these, the road infrastructure can be changed by policymakers to reduce the severity of accidents.

Reynolds et al. [19] reviewed studies on the impacts of road infrastructure on cyclist safety. Intersection studies have found that multilane roundabouts can significantly increase the risk to cyclists unless a separated cycle track is included in the design. Other studies have suggested that sidewalks and multi-use trails pose the highest risk, that major roads are more hazardous than minor roads, and that the presence of cyclist facilities is associated with the lowest risk. Helak et al. [20] suggested that simply providing a dedicated space for cyclists, such as a bike lane or a paved shoulder, did not reduce the severity of the sustained

injuries in a crash with a motor vehicle. Goerke et al. [21] showed that increasing the miles of bike lanes was associated with a significant reduction in severe head injuries; moreover, the miles of bike lanes were not associated with any significant changes in mortality or mechanical ventilation days when adjusted for other factors.

The factors associated with the severity of cyclist crashes may differ across different cyclist crash patterns. By considering the systematic heterogeneity of cyclist crashes, researchers and practitioners can identify the most appropriate safety countermeasures for different cyclist crash types under different conditions. One way to account for the heterogeneous nature of this data is to use a latent class clustering approach. A latent class analysis can segment the cyclist crash data into mutually exclusive and exhaustive latent classes by assuming latent categorical variables. The class membership of each cyclist crash can then be inferred from the observed variables.

Kaplan and Prato [22] analyzed the patterns of cyclist–motorist crashes in Denmark and investigated their prevalence and severity. They obtained 13 distinguishable cyclist–motorist classes using a latent class clustering approach. The prevalent features allowing for differentiation of the latent classes were the speed limit, infrastructure type, road surface conditions, number of lanes, motorized vehicle precrash maneuvers, availability of cycle lanes, cyclist intoxication, and helmet-wearing behavior. Based on an analysis of these 13 cyclist crash patterns, three types of safety considerations were proposed. These safety considerations were related to network design and connectivity, road maintenance, and cyclist road behavior.

Prati et al. [23] identified subgroups of cyclist crashes in Italy and analyzed the different cyclist crash-type subgroups separately. Using a latent class analysis, a cyclist crash dataset was segmented into 19 classes. A logistic regression analysis was used to identify the associations between each class membership and the severity of the cyclist crashes. Finally, association rules were established for each latent class to uncover the factors associated with an increased likelihood of severity.

Myhrmann et al. [24] explored the factors related to the injury severity outcomes in single-cyclist crashes. Three latent classes were identified using a latent class-ordered probit model. They found that the likelihood of cyclist membership in such classes depended on the cyclist's age and gender. Furthermore, the road geometry, maintenance level, and interactions between the road geometry and maintenance level affected the severity.

Samerei et al. [25] identified factors affecting motor vehicle–bicycle crashes in Victoria, Australia using a binary logit model and latent class clustering. They found the factors that increased the risk of fatalities and serious injuries of cyclists in all the clusters were being an elderly cyclist, not using a helmet, and dark conditions.

Liu et al. [26] analyzed the impacts of various factors on cyclist injury severity at intersection and non-intersection locations using a latent class clustering analysis and mixed logit models. The factors with significant impacts on cyclist injury severity at intersections included cyclists drinking alcohol; driving a van, bus, or single-unit truck; motorist fault; inclement weather; and dusk or dawn conditions. In contrast, those with significant impacts on cyclist injury severity at non-intersection locations included the cyclist gender, drivers drinking alcohol, vehicle speed, speeding, type of area (rural or urban areas), traffic control, and curved roads.

Despite the above, relatively few studies have used such crash classifications, and it is not clear which factors contribute the most to these classifications. In Japan, where the number of cyclist fatalities was the highest among all the Organization for Economic Co-operation and Development countries in 2009 (see. https://www.cyclehelmets.org/1258.html (accessed on 21 March 2022).), latent class analyses have not been conducted owing to data limitations. There have been analyses of each factor and its severity [27,28], but it remains unclear which factors are the most associated with a fatality risk. The impacts of various road infrastructure elements also remain unclear.

In addition, relatively few studies have examined and compared the different factors contributing to cyclist injury severity at different locations [26]. Furthermore, a fatality

accident is a rare event; thus, ordinary logistic regression is inappropriate. Even with a large number of observations, the benefits of the loss functions that normal machine learning algorithms work with may be minimized, e.g., by the machine learning predicting that all of the inputs are of the majority class. The stronger the imbalance of the outcome, the more severe the bias in the predicted probabilities. In a previous study, a penalized maximum likelihood estimation was applied to cancel out this bias [29].

This study focused on cyclist crashes occurring in Japan between 2019 and 2020. To identify reliable and relevant subgroups of cyclist crashes, we used a polytomous latent class analysis. The polytomous manifest variables included categorical indicators corresponding to characteristics of the road infrastructure, cyclist(s), vehicle(s), and environmental conditions. Specifically, characteristics concerning the infrastructure (road type, stop sign, and crash location); cyclist (age of the cyclist and collision part); collision partner (trucks and obstacles); environment (surface conditions and weather); and time period were employed in the analysis, following Prati et al. [23]. Notably, data concerning cyclist injury severity were not included in the latent class analysis, as the cyclist injury severity is considered as an outcome of a crash. We showed that the crash location (intersection, single road, and other factors) contributes the most to the classification. Then, based on crash location, we investigated the cyclist injury severity (e.g., fatality or not) using a bias-reduced logistic regression analysis. This analysis allowed us to identify the factors significantly affecting cyclist crashes.

## 2. Materials and Methods

The crash data for the models were provided by the National Police Agency of Japan (NPA). The original database comprised 690,415 road accidents occurring from 2019 to 2020 on Japanese roads. The NPA database includes all the publicly available data for accidents in Japan but does not include data prior to 2019.

To narrow down the events to those pertinent to the current study, we extracted 139,955 accidents in which at least one cyclist was injured or killed. The NPA database distinguishes between road accidents resulting in injuries or fatalities (within 30 days); however, it does not distinguish between different levels of injury. Table 1 shows the descriptive statistics of the 15 categorical variables selected for this dataset: (1) accident location type, (2) stop sign, (3) restricted speed, (4) median strip, (5) boundary between sidewalk and roadway, (6) road type, (7) road alignment, (8) land uses, (9) surface conditions, (10) weather, (11) time period, (12) collision partner, (13) cyclist's age, (14) collision part of the bicycle, and (15) severity of the cyclist crash. According to the Japanese Road Code, there are five types of roads: national roads, trunk roads managed by prefecture and municipality, and prefectural and municipal roads. Unfortunately, the speed of the motor vehicle involved in the incident, speed of the cyclist, and violation of the rules were not included in the dataset.

There are more than 9.1 billion possible combinations of such accidents, but in actuality, 90,698 combinations are observed. There are two levels of severity, crashes with and without fatalities. There are 5917 (0.9%) crashes with fatalities in all the accidents, and 807 (0.6%) in cyclist-related accidents. The fatality rate in cyclist-related accidents is lower than that in all the accidents; this is because pedestrians, which represent the most vulnerable group, are not the focus of this research.

These variables and their distributions are shown in Table 1. In relation to land use, the term DIDs refers to densely inhabited districts, which are a series of census districts with a population density of 4000 or more per square kilometer and a population of more than 5000.

**Table 1.** Variables and their distributions.

| Variable | Category | n | % |
|---|---|---|---|
| Crash location | Between intersections | 43,337 | 31.0 |
| | Intersections | 93,367 | 66.7 |
| | Sidewalks | 3251 | 2.3 |
| Stop sign | Not applicable * | 47,866 | 34.2 |
| | YES | 22,672 | 16.2 |
| | NO | 69,417 | 49.6 |
| Restricted speed | 20 | 2573 | 1.8 |
| | 30 | 20,967 | 15.0 |
| | 40 | 32,888 | 23.5 |
| | 50 | 16,107 | 11.5 |
| | 60 | 69,420 | 49.6 |
| Median strip | Yes | 16,452 | 11.8 |
| | High-brightness paint | 170 | 0.1 |
| | Chatter bar | 453 | 0.3 |
| | Postcorn | 495 | 0.4 |
| | Paint | 55,093 | 39.4 |
| | None | 67,292 | 48.1 |
| Boundary between sidewalk and roadway | Guard rail | 12,015 | 8.6 |
| | Curb | 80,437 | 57.5 |
| | White line | 23,959 | 17.1 |
| | None | 23,544 | 16.8 |
| Road type | National | 19,113 | 13.7 |
| | Trunk- Prefectural | 19,222 | 13.7 |
| | Trunk- Municipal | 1535 | 1.1 |
| | Prefectural | 14,184 | 10.1 |
| | Municipal | 85,901 | 61.4 |
| Alignment | Right curve - up | 292 | 0.2 |
| | Right curve - down | 313 | 0.2 |
| | Right curve - flat | 1441 | 1.0 |
| | Left curve - up | 227 | 0.2 |
| | Left curve - down | 286 | 0.2 |
| | Left curve - flat | 1213 | 0.9 |
| | Straight - down | 2667 | 1.9 |
| | Straight - flat | 4694 | 3.4 |
| | Straight - up | 128,819 | 92.0 |
| Land uses | (ref) Urban - DID | 86,535 | 61.8 |
| | Urban - non DID | 38,523 | 27.5 |
| | Rural | 14,897 | 10.6 |
| Surface conditions | Dry | 124,271 | 88.8 |
| | Wet | 15,506 | 11.1 |
| | Frozen | 69 | 0.0 |
| | Snow cover | 39 | 0.0 |
| | Unpaved | 70 | 0.1 |
| Weather | Clear | 99,374 | 71.0 |
| | Cloudy | 27,269 | 19.5 |
| | Rainy | 13,083 | 9.3 |
| | Foggy | 28 | 0.0 |
| | Snowy | 201 | 0.1 |
| Time period | After dawn | 4353 | 3.1 |
| | Daytime | 93,833 | 67.0 |
| | Before sunset | 10,086 | 7.2 |
| | After sunset | 10,471 | 7.5 |
| | Nighttime | 19,769 | 14.1 |
| | Before dawn | 1443 | 1.0 |

**Table 1.** *Cont.*

| Variable | Category | n | % |
|---|---|---|---|
| Collision partner | Cars | 64,697 | 46.2 |
| | Kei-cars | 31,527 | 22.5 |
| | Large truck | 4643 | 3.3 |
| | Small/Medium truck | 16,495 | 11.8 |
| | Motorcycle 126+cc | 1134 | 0.8 |
| | Motorcycle -125cc | 5735 | 4.1 |
| | Bicycle | 5438 | 3.9 |
| | Pedestrian | 5140 | 3.7 |
| | Obstacles | 794 | 0.6 |
| | None | 4352 | 3.1 |
| Cyclist's age | 0–24 | 48,602 | 34.7 |
| | 25–34 | 17,230 | 12.3 |
| | 35–44 | 16,598 | 11.9 |
| | 45–54 | 17,090 | 12.2 |
| | 55–64 | 12,188 | 8.7 |
| | 65–74 | 13,893 | 9.9 |
| | 75– | 14,354 | 10.3 |
| Collision part | None | 2785 | 2.0 |
| | Front | 56,729 | 40.5 |
| | Right | 43,647 | 31.2 |
| | Rear | 4979 | 3.6 |
| | Left | 31,815 | 22.7 |
| Accident type | Injury | 139,148 | 99.4 |
| | Fatality | 807 | 0.6 |

Note: * Collision partner is not motor vehicles.

*Methods*

To perform the polytomous latent class analysis, we used the poLCA package in R software [30,31]. The latent class model estimates the observed joint distribution of the categorical variables in a set of observations and approximates it as a weighted sum of a finite number $R$, which is based on a large multiway contingency table.

In this study, we selected the number of classes using the Bayesian information criterion (BIC). In general, the rule of "the smaller, the better" should be applied when evaluating fit indices. To ensure that the global maximum likelihood of the latent class model was achieved (rather than the local maximum log-likelihood), we re-estimated each model 5000 times and set 10 initial values of the class-conditional response probabilities; then, we saved the model with the greatest likelihood.

From the analysis, we could determine the amount of each category of each variable included in each class. Based on the chi-square goodness-of-fit tests and tests of the differences between the proportions, we ascertained the characteristics of each class based on whether the proportion was significantly higher (or lower) than the overall proportion ($p < 0.001$).

In this study, the most important variable contributing to the classification was the crash location (e.g., intersections, areas between intersections, and sidewalks). Therefore, we divided the data by crash location and analyzed the risk of death. However, as mentioned above, the risk of death was generally very small. Therefore, we employed Firth's bias-reduced logistic regression to perform the fitting using pseudo-data representations [29]. To take advantage of bias-reduced logistic regression, we used an R package called brglm [32]. The bias-reduced estimator is second-order unbiased and has a smaller variance than the maximum likelihood estimator. Using the backwards stepwise method, we estimated the model that minimized the Akaike information criterion and analyzed the risk factors in fatality accidents.

## 3. Results

### 3.1. Polytomous Latent Clustering Analysis

#### 3.1.1. Selection of Number of Classes

By increasing the number of class settings from two, the optimal number of classes with the smallest BIC value was obtained. As shown in Figure 1, poLCA software yielded 17 latent classes.

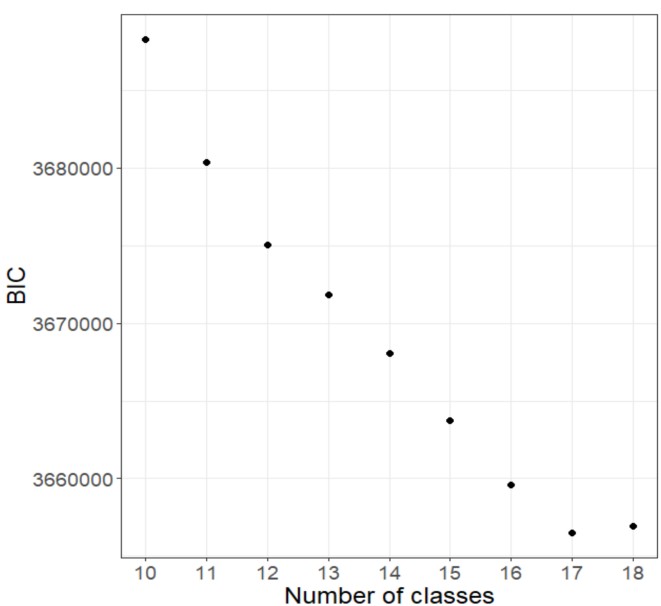

**Figure 1.** Change in the Bayesian information criterion (BIC) as a function of the number of latent classes.

#### 3.1.2. Estimation Results

We tested the differences between the proportions in the full sample and those in the class. The categories showing statistical significance for each variable are depicted in Figure 2.

As an initial matter, the crash location was detected in all classes: C1–8 were intersections (66.7%), C9–14 were between intersections (31.0%), and C15–17 were between intersections and sidewalks (2.3%). For each of these three locations, we renumbered the classes in descending order of the fatality accident rates in each class. Vertical items were classified according to the road structure, environment, driver, and cyclist status. The categories for which significant differences were found for each variable are shown. If the categories were spread across multiple classes, they were bundled together. C17 had the highest fatality rate, followed by C14, 13, and 12. The fatality rate for C17 was 3.1%, i.e., more than five times the fatality rate for the entire sample. The characteristics of each variable are as follows.

Among C1–8, where the stop sign variable appeared as a feature within the intersections, "NO" appeared as a feature in C1 and C5–8. The percentages of the observations by class for "NO" were 49.6% (total sample), 70.5% (C1), and 82.8%, 84.6%, 98.4%, and 86.8% (C5–8), respectively.

The restricted speed varied considerably, with "30 km/h" for C2–3, "40 km/h" for C9 and 14, "50 km/h" for C6 and 8, "40 km/h" and "50 km/h" for C7, "60 km/h" for C4–5 and 15, and "50 km/h" and "60 km/h" for C11. The percentages of the observations by class for each restricted speed were 15.0% (overall), 49.0% (C2), and 29.6% (C3) for "30 km/h", 23.5% (overall), 73.3% (C7), 47.6% (C9), and 52.1% (C14) for "40 km/h"; 11.5% (overall), 26.1%, 25.8%, 37.9% (C6–8), and 35.5% (C11) for "50 km/h"; and 36.9% (overall), 50.4% (C4), 61.3% (C5), 56.5% (C11), and 58.6% (C15) for "60 km/h".

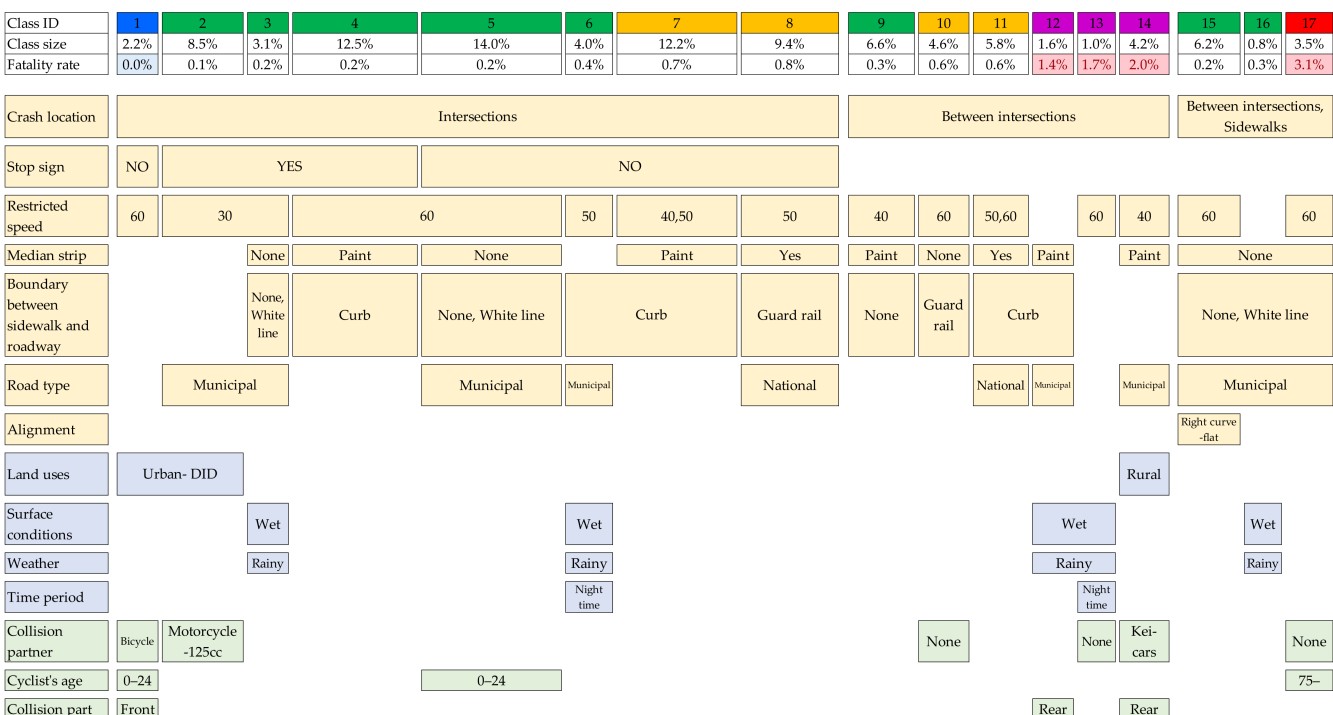

**Figure 2.** Characteristics of the 17 classes.

Regarding a median strip, C3, 5, 10, and 15–17 were characterized by "None"; C4, 7, 9, 12, and 14 were characterized by "Paint"; and C8 and 11 were characterized by "Yes". C10 was represented as a feature on the side, i.e., with less than the proportion of the total sample. The percentages of the observations by class for each median strip were 48.1% (overall), 77.1% (C3), 96.6% (C5), 20.8% (C10), 93.9%, 84.2%, and 90.5% (C15–17) for "None"; 39.4% (overall), 59.8% (C4), 62.6% (C7), 79.0% (C9), and 91.1% (C14) for "Paint"; and 11.8% (overall), 49.9% (C8), and 48.6% (C11) for "Yes".

On the boundary between the sidewalk and roadway, C8 and 10 were characterized by "Guard rail"; C4, 6, 7, 11, and 12 were characterized by "Curb"; C3, 5, and 15–17 were characterized by "None, white line"; and C9 was characterized by "None". C9 was represented as a feature on the side, i.e., with less than the proportion of the total sample. The percentages of the observations by class for each boundary between the sidewalk and roadway were 8.6% (overall), 21.5% (C8), and 23.1% (C10) for "Guard rail"; 57.5% (overall), 71.1% (C4), 82.0% (C6), 81.6% (C7), 82.9% (C11), and 81.6% (C12) for "Curb"; 17.1% (overall), 32.3% (C3), 33.3% (C5), 34.1%, 31.6%, and 32.0% (C15–17) for "White line"' and 16.8% (overall) and 1.7% (C9) for "None".

Regarding the characteristics of the road type, "National road" was detected in C11; "National and trunk-municipal road" was detected in C8; and "Municipal road" was detected in C2, 3, 5, 6, 12, and 14–17. C6, 12, and 14 were represented as a feature on the side with less than the proportion of the total sample. The percentages of the observations by class for each road type were 13.7% (overall), 47.8% (C8), and 41.6% (C11) for national road; 1.1% (overall) and 5.8% (C8) for "Trunk municipal road"; and 61.4% (overall), 86.0% (C2), 90.2% (C3), 96.1% (C5), 33.8% (C6), 40.1% (C12), 36.2%, 95.3%, 93.0%, and 95.6% (C14–17) for "Municipal road".

Regarding alignment, only one feature was expressed, i.e., "Right curve flat" at C15. The percentages of the observations by class were 1.0% (overall) and 5.2% (C15) for the right curve flat.

The characteristic of the "Urban DID (Densely Inhabitant District) area" was classified into C1 and C2, whereas that in the "Rural area" was classified in C14. The percentages of the observations by class for each land use were 61.8% (overall), 83.2% (C1), and 87.8% (C2) for "Urban-DID" and 10.6% (overall) and 32.2% (C14) for "Rural". The land uses

were clearly characterized, with low-fatality values at C1 and 2 and high values at C14, suggesting that the fatality risk may be higher in rural areas.

Regarding the surface conditions, C3, 6, 12, 13, and 16 showed "Wet" features. The percentages of observations by class for "Wet" were 11.1% (overall), 97.5% (C3), 97.3% (C6), 97.8% (C12), 99.4% (C13), and 99.3% (C16). Regarding weather, C3, 6, 12, 13, and 16 had "Rainy" features, i.e., the same classes in which wet was expressed on the road surface. The percentages of observations by class for "Rainy" were 9.4% (overall), 83.2% (C3), 84.3% (C6), 82.8% (C12), 79.3% (C13), and 79.1% (C16).

The time period was detected only in C6 and 13, as "Nighttime". The percentages of observations by class for "Nighttime" were 14.1% (overall), 30.0% (C6), and 29.0% (C13).

The characteristics of the collision partner were estimated as only "Bicycle" for C1; only "Motorcycle -125cc" for C2; "None" for C10, 13, and 17; and only "Kei-cars" for C14. The percentages of the observations by class for each collision partner were 3.9% (overall) and 100.0% (C1) for "Bicycle"; 4.1% (overall) and 11.1% (C2) for "Motorcycle -125cc"; and 3.1% (overall), 26.1% (C10), 43.4% (C13), and 42.8% (C17) for "None". Hence, C1 consisted entirely of cyclist accidents. C2–8, 9, 11–12, and 14–16 consisted of "(Kei-)cars, Truck, and Motorcycles" as the collision partners, whereas C10, 13, and 17 consisted of "Bicycle, Pedestrian, Obstacles, and None".

Similar to previous research, the cyclist's ages were detected in some classes. C1 and 5 showed "0–24", and C17 showed "75–". C1 was represented as a feature on the side with less than the proportion of the total sample. The percentages of the observations by class for each cyclist year were 34.7% (overall), 13.5% (C1), and 51.9% (C5) for "0–24" and 10.3% (overall) and 21.1% (C17) for "75–". Age "75–" was shown as a feature only in C17 and had the highest fatality rate, suggesting that it may be a factor increasing the fatality risk.

Finally, in the collision part, C1 was indicated by "Front", and C12 and 14 were indicated by "Rear". The percentages of the collision part were 40.5% (overall) and 64.6% (C1) for "Front" and 3.6% (overall), 11.1% (C12), and 12.0% (C14) for "Rear".

### 3.2. Bias-Reduced Logistic Regression

As the polytomous latent class analysis revealed that the variable contributing the most to the classification was the crash location, a bias-reduced logistic regression analysis was conducted for each crash location. Table 2 presents the estimated results. In the following section, only those judged to be statistically significant at the 5% significance level are mentioned. The asterisks *, **, and *** indicate that the coefficients are statistically different from zero at the 5%, 1%, and 0.1% levels, respectively.

In all crash locations, older age and "Rural areas" are identified as common factors increasing the fatality risk. In addition, "No stop sign" at an intersection is found to increase the fatality risk. According to this result, it is believed that the installation of stop signs has some effect on reducing fatality accidents, although such signs not being applicable may also increase the fatality risk. However, this is possibly because the majority of collision partners are pedestrians and because the cyclist is alone during the accident.

Regarding the road type, "Trunk prefectural road" and "Municipal road" have low risks of fatality accidents within and between intersections. In contrast, "Prefectural Road" is found to have a low risk of fatality at intersections but a high risk of fatality on sidewalks. In addition, the fatality risk on sidewalks is also higher for "Prefectural road".

Regarding the median strip, "Paint and None" increase the risk of fatality accidents both within and between intersections. "Chatter bar" increases the fatality risk only at intersections. "High-brightness paint" decreases the fatality risk at intersections but, conversely, increases the fatality risk between intersections. However, there were only two fatalities at intersections with the "Chatter bar", and only one fatality was observed between intersections with "High-brightness paint". Therefore, it is possible that these results were obtained only by chance, and a more detailed analysis is required in the future.

**Table 2.** Estimation results.

| Crash Location (Fatality Rate) | | Intersections (0.4%) | | | | | Between Intersections (0.8%) | | | | | Sidewalks (3.0%) | | | | |
|---|---|---|---|---|---|---|---|---|---|---|---|---|---|---|---|---|
| | Variable | Coef. | z Value | | Fatality | n | Coef. | z Value | | Fatality | n | Coef. | z Value | | Fatality | n |
| Cyclist's age | (ref) 0–24 | | | | 47 | 34,898 | | | | 12 | 12,846 | | | | 1 | 858 |
| | 25–34 | −0.42 | −1.20 | | 9 | 11,360 | 0.15 | 0.31 | | 6 | 5490 | 2.50 | 2.43 | * | 3 | 380 |
| | 35–44 | −0.45 | −1.22 | | 8 | 10,629 | 1.20 | 3.33 | *** | 18 | 5510 | 2.85 | 2.80 | ** | 3 | 418 |
| | 45–54 | 0.49 | 1.92 | | 22 | 10,832 | 1.69 | 5.15 | *** | 34 | 5820 | 1.90 | 1.99 | * | 5 | 438 |
| | 55–64 | 1.29 | 5.85 | *** | 36 | 7817 | 1.86 | 5.56 | *** | 29 | 4035 | 3.19 | 3.57 | *** | 15 | 336 |
| | 65–74 | 1.90 | 10.37 | *** | 79 | 8923 | 2.84 | 9.40 | *** | 80 | 4573 | 3.42 | 3.96 | *** | 29 | 397 |
| | 75– | **2.71** | 16.61 | *** | 178 | 8908 | **3.39** | 11.56 | *** | 150 | 5022 | **3.43** | 4.01 | *** | 43 | 424 |
| Land uses | (ref) Urban-DID | | | | 185 | 56,928 | | | | 120 | 27,671 | | | | 24 | 1936 |
| | Urban-non DID | 0.34 | 2.77 | ** | 112 | 26,813 | 0.57 | 3.91 | *** | 87 | 10,772 | 1.16 | 3.50 | *** | 25 | 938 |
| | Rural | **1.04** | 7.61 | *** | 82 | 9626 | **1.36** | 10.03 | *** | 122 | 4894 | **1.87** | 6.23 | *** | 50 | 377 |
| Road type | (ref) National | | | | 98 | 12,724 | | | | 83 | 6152 | | | | 1 | 237 |
| | Trunk-Prefectural | −0.48 | −2.88 | ** | 57 | 12,340 | −0.36 | −2.04 | * | 59 | 6497 | −0.06 | −0.05 | | 1 | 385 |
| | Trunk-Municipal | −0.20 | −0.46 | | 5 | 1087 | 0.80 | 1.61 | | 4 | 425 | 0.42 | 0.24 | | - | 23 |
| | Prefectural | −0.39 | −2.17 | * | 48 | 9083 | −0.30 | −1.61 | | 51 | 4782 | **2.27** | 2.30 | * | 14 | 319 |
| | Municipal | **−0.81** | −5.81 | *** | 171 | 58,133 | **−0.95** | −5.79 | *** | 132 | 25,481 | 1.47 | 1.59 | | 83 | 2287 |
| Median strip | (ref) Yes | | | | 32 | 10,279 | | | | 31 | 5887 | | | | | |
| | High-brightness paint | −0.44 | −6.91 | *** | | 109 | **1.87** | 2.12 | * | 1 | 59 | | | | | |
| | Chatter bar | **1.41** | 2.12 | * | 2 | 228 | 1.00 | 1.67 | . | 3 | 224 | | | | | |
| | Postcorn | 1.12 | 1.68 | . | 2 | 306 | −0.60 | −0.42 | | - | 183 | | | | | |
| | Paint | 0.53 | 2.65 | ** | 135 | 32,958 | 0.57 | 2.83 | ** | 198 | 21,113 | | | | | |
| | None | 0.87 | 4.36 | *** | 208 | 49,487 | 0.46 | 1.98 | * | 96 | 15,871 | | | | | |
| Stop sign | (ref) Yes | | | | 23 | 22,507 | | | | | | | | | | |
| | No | 1.41 | 6.63 | *** | 336 | 68,926 | | | | | | | | | | |
| | Not applicable * | **2.10** | 6.90 | *** | 20 | 1934 | | | | | | | | | | |

**Table 2.** *Cont.*

| Crash Location (Fatality Rate) | | Intersections (0.4%) | | | | | Between Intersections (0.8%) | | | | | Sidewalks (3.0%) | | | | |
|---|---|---|---|---|---|---|---|---|---|---|---|---|---|---|---|---|
| | Variable | Coef. | z Value | | Fatality | n | Coef. | z Value | | Fatality | n | Coef. | z Value | | Fatality | n |
| **Boundary between sidewalk and roadway** | (ref) Guard rail | | | | 72 | 7250 | | | | | | | | | | |
| | Curb | −1.03 | −7.37 | *** | 219 | 53,166 | | | | | | | | | | |
| | White line | −1.22 | −6.14 | *** | 48 | 16,390 | | | | | | | | | | |
| | None | **−1.58** | −7.28 | *** | 39 | 16,561 | | | | | | | | | | |
| **Collision partner** | (ref) Bicycle | | | | | | | | | 2 | 2274 | | | | | |
| | Cars | | | | | | 1.40 | 2.20 | * | 76 | 16,986 | | | | | |
| | Kei-cars | | | | | | 1.39 | 2.15 | * | 44 | 7964 | | | | | |
| | Large truck | | | | | | **2.97** | 4.63 | *** | 61 | 1736 | | | | | |
| | Small/Medium truck | | | | | | 1.97 | 3.06 | ** | 44 | 4683 | | | | | |
| | Motorcycle 126+cc | | | | | | 2.40 | 3.21 | ** | 6 | 375 | | | | | |
| | Motorcycle -125cc | | | | | | 0.36 | 0.44 | | 3 | 1736 | | | | | |
| | Pedestrian | | | | | | 0.52 | 0.67 | | 4 | 3743 | | | | | |
| | Obstacles | | | | | | 2.80 | 4.16 | *** | 17 | 629 | | | | | |
| | None | | | | | | 2.47 | 3.88 | *** | 72 | 3185 | | | | | |
| **Time period** | (ref) Except for nighttime | | | | | | | | | 211 | 36,687 | | | | | |
| | Nighttime | | | | | | **1.45** | 11.78 | *** | 118 | 6650 | | | | | |
| **Crach part** | (ref) Except for front | | | | | | | | | | | | | | 42 | 1875 |
| | Front | | | | | | | | | | | **1.26** | 4.91 | *** | 57 | 1376 |
| **Intercept** | | −7.22 | −22.16 | *** | | | −9.29 | 13.06 | *** | | | −11.57 | −8.53 | *** | | |
| AIC | | | | 4210 | | | | | 3004 | | | | | 442 | | |
| AIC (null) | | | | 4933 | | | | | 3869 | | | | | 888 | | |
| Number of samples | | | | 93,367 | | | | | 43,337 | | | | | 3251 | | |

The asterisks *, **, and *** indicate that the coefficients are statistically different from zero at the 5%, 1%, and 0.1% levels, respectively.

At the intersection, the boundary between the sidewalk and roadway affects the severity of the injury. The fatality risk is higher when a "Guard rail" is installed. This result may be owing to the fact that cyclists cannot "escape" to the sidewalk when they are riding on the side of the roadway on a road where a guard rail is installed, even if they are in danger of colliding with a car.

The collision partner is an important factor for severity but only between intersections. The fatality risk is found to be high in all cases, except for when the collision partner was a "Bicycle, Pedestrian, or Motorcycle −125cc", and the fatality risk was particularly high for "Large track, Obstacles, and Motorcycle 126+cc". These results suggest that the intensity of the impact at the time of the accident has a significant effect on the risk of death.

Other findings include that the fatality risk is higher at "Nighttime" between intersections and when the collision part is the "Front" on sidewalks. Since vehicle recognition is delayed at "Nighttime" owing to darkness, both of these factors can be attributed to the inattention of the cyclist.

## 4. Discussion

### 4.1. Accident Classification

The results of the polytomous latent class analysis show that cyclist accidents in Japan can be classified into 17 classes. In previous studies, there were 13 cases found in Denmark and 19 in Italy [22,23]. Although the number of variables and the categories addressed are different among these studies, approximately 10,000–70,000 cyclist accidents were classified into less than 20 classes based on the BIC.

The most important contributing factor to the classification in this study is the crash location. Interestingly, this result is the same as that obtained in Italy. In contrast, in Denmark, the area (urban or rural) was the first dividing factor, and crash location was the third factor. In Japan, the area is also one of the factors for classification but is generally a more important factor for explaining the severity. In Italy, the area was not included as a variable. In Denmark and Italy, the rule is for drivers to drive on the right side of the road, whereas, in Japan, they drive on the left. However, this rule had no impact on the classification.

### 4.2. Intersection Accidents

The bias-reduced logistic regression analysis revealed that the factors increasing the fatality risk for cyclist accidents at intersections were "older cyclist", "Rural area", "National road", "Chatter bar", "Paint", and "None" in the median strip and "No stop sign" and "Guard rail" in the boundary between the sidewalk and roadway. "High-brightness paint" in the median strip is also shown to be a factor reducing the fatality risk. However, the effectiveness of "High-brightness paint" in the median strip remains somewhat uncertain, as there are only a few sections where this system has been introduced, and only 109 corresponding traffic accidents were confirmed.

Shen et al. [14] found that urban junctions had a higher fatality risk for cyclists, although this study found that the fatality risk was lower in urban areas than in rural areas.

The installation of stop signs and median separations may reduce the risk of fatalities and should be promoted in the same manner as in other countries. However, the existence of a guard rail is another factor increasing the fatality risk. In Japan, cyclists often ride on sidewalks or the wrong way [33]. It is possible that the cyclists riding on sidewalks may be induced to do so by the guard rail. Reynolds et al. [19] mentioned that the existing research suggests that most studies on sidewalk-riding found that it is a very dangerous behavior for cyclists. The excessive segregation of vulnerable road users and motor vehicles can create risks. For example, roadside barriers can encourage motor vehicle drivers to reduce their attentiveness toward other road users.

It is important to reduce the complexity of conflict points. For example, grade-separated junctions segregate primary traffic streams by placing them on separate levels; this measure is used for roads with high traffic volumes. Other treatments to redesign

junctions include changes in the angles between roads and/or measures to improve the lines of sight and eliminate objects obscuring approaches. The concept of "daylighting" refers to the removal of parking spaces in front of curbs around an intersection, thereby increasing the visibility for pedestrians and drivers. In this study, the median strip was a factor that both increased and decreased the fatality risk. However, the sample size used herein may be insufficient. In addition, our data do not reflect physical intersection structures such as normal crossroads, T-junctions, and roundabouts. Accordingly, further investigation is required.

*4.3. Non-Intersection Accidents*

The bias-reduced logistic regression analysis revealed that the factors increasing the fatality risks of cyclist accidents between intersections were "older cyclist", "Rural area", "National road", "Paint", and "None" and, in the median strip, "Trucks as the collision partner" and "Nighttime". The factors identified for sidewalks include "older cyclists", "Rural area", and "Prefectural road".

The increased risk of fatalities at night has been reported in several recent studies [21,25,26,34]. This higher risk of fatalities is owing to a variety of factors, such as delays in detecting hazards at nighttime owing to poor visibility relative to daytime and higher speeds of vehicles owing to lower traffic volumes.

Asgarzadeh et al. [20] showed that collisions with large vehicles generally increase the fatality risk in cyclist accidents; this result was also identified in this study as a factor that increased the fatality risk in cyclist accidents between intersections. This study also found that collisions with "Obstacles" and "Motorcycle 126+cc" also significantly increased the fatality risk. In collisions with "Obstacles", the cyclist's inattention to decelerate immediately before an accident may be a contributing factor to the strong impact of collisions, whereas, in collisions with "Motorcycle 126+cc", the motorcyclist's high acceleration may make accidents more likely to occur at high speeds.

The higher fatality risk found for cyclist accidents in non-intersection rural areas identified by Liu et al. [26] was also shared by this study. Liu et al. [27] found that the fatality risk of cyclist accidents was also high in urban areas. This study showed that the fatality risk was higher in a rural area, "Urban-non-DID area", and "Urban-DID area" in that order and identified areas where measures were particularly needed to effectively reduce the fatality risk of cyclist accidents.

The fatality risk on sidewalks is higher than that on prefectural roads. Of the 319 accidents, 14 were fatal, indicating a high fatality risk. The reason for this may be that the corresponding surfaces are less well-maintained than those of national and municipal roads. Since one of the parts of a bicycle involved in collisions is the front, it is conceivable that the holes in such sidewalks could lead to serious accidents. Uneven roads may exist as a result of insufficient road maintenance. Therefore, promoting these improvements may reduce the risk of death. Unfortunately, we did not collect data regarding the surface pavement conditions of the sidewalks and roadways. Thus, further research is necessary.

Finally, at both intersections and non-intersections, the restricted speed, weather, and surface conditions did not affect the severity of cyclist accidents. This result is different from that of previous studies [4,11]. However, the reasons for this are unclear, owing to the impossibility or inability to obtain detailed or specific data for such analyses.

## 5. Conclusions

In this study, we analyzed and classified cyclist crashes occurring in Japan between 2019 and 2020. We employed a latent class analysis using categorical indicators related to the infrastructure, road users, vehicles, and environmental/time characteristics to classify cyclist crashes. We identified 17 classes of cyclist crashes with different causes, degrees of severity, and profiles of road users. The variable contributing the most to the classification was the crash location. Common fatality risks included older age groups and rural areas, whereas other factors differed among the crash locations. The existence of median strips,

stop signs, and boundaries between sidewalk and roadways affected crashes at intersections. The median strip, collision partner, and time period affected crashes between intersection. On the sidewalks, the fatality risk was higher when the collision part of the bicycle was the front.

The results of this study suggest that it is important to implement countermeasures based on the crash location. At intersections, the installation of stop signs and median separations may reduce the risk of fatalities. It is common for cyclists to decelerate when approaching a stop sign and check to ensure that there is no vehicular or pedestrian cross-traffic before proceeding through the intersection rather than putting a foot down. Many different parties can be liable for a bike accident at a stop sign, including the cyclist.

This study showed that a median separation was one of the factors reducing the severity, and the installation of guard rails potentially increases the risk of death. Wang et al. showed that tree separation for non-motorists increased the injury severity of e-bicyclists [35]. It would be worthwhile to investigate other separation treatments that could better protect e-bicyclists from crashes.

The presence of bumps and obstacles, particularly on rural roads with low pedestrian traffic, has rarely been reported to road managers, which may lead to an increase in single-vehicle accidents. Regular road maintenance of the shoulders and other non-roadway sections is important.

However, this study has the following issues regarding the data and methods. The first challenge regarding the data limitations is the definition of severity. In this study, only two classifications were used: fatal or non-fatal. Fatalities are extremely rare, and future analyses by subdividing the injuries such as hospitalization or not may yield different results. In addition, the data collected in this study do not include detailed attributes of cyclists. Previous studies have analyzed the factors such as alcohol consumption, helmet use, and improper behaviors such as violations of the law [4,5,11]. It is important to take these factors into account in the analysis of the fatality risk. Furthermore, the lack of traffic flow data for motor vehicles, cyclists, and pedestrians is a major deficiency. For example, the installation or removal of guard rails and utility poles should be considered not only for cyclists, but also for pedestrians and motor vehicles. In Japan, there are many cases of cyclists running on the wrong side of sidewalks and roadways and automobiles parking in bicycle lanes. On urban roads without a bike lane, the vehicle speeds with and without cyclists were found to be negligible [36]. On-road bicycle lanes and parked cars also reduced the passing distance [37]. It is desirable to analyze these infrastructures on the severity combined with the mean and variances traffic volume by various road users and land use along roadsides data.

Regarding the methodology, latent class analysis provides the best fit when modeling the data but does not provide an indication of how well the clusters are separated. The popularity of such methods might be explained by the apparent implication that the results show clear (i.e., separate) clusters, which are then assumed to represent a qualitatively distinct group of people with an existing disease entity. However, it should be noted that the classes formed are relative to each other and do not necessarily have an inherent meaning.

Finally, Billot-Grasset et al. [15] pointed out that road user behavior influences each step in the chain of events leading to an accident. Ma et al. [16] showed that cyclists' crash risks were directly predicted by risky cycling behaviors and cycling anger and that the effects of cycling anger, impulsiveness, and normlessness on crash risks were mediated by cycling behaviors. These psychological aspects of cyclists and drivers are important for understanding accidents. The development of bicycle spaces will ultimately lead to the creation of safe travel spaces for all road users [38]. Although underreporting biases has been noted [39], not simply the severity but also the probability of accidents need to be examined further.

**Author Contributions:** Conceptualization, Y.S. and M.T.; methodology, Y.S. and M.T.; software, Y.S and M.T.; validation, M.T. and D.S.; formal analysis, Y.S. and M.T.; investigation, Y.S.; resources, M.T.; data curation, Y.S.; writing—original draft preparation, Y.S. and M.T.; writing—review and editing, Y.S. and M.T.; visualization, Y.S. and M.T.; supervision, M.T.; and project administration, M.T. All authors have read and agreed to the published version of the manuscript.

**Funding:** This study received no external funding.

**Data Availability Statement:** Publicly available datasets were analyzed in this study. These data are available at https://www.npa.go.jp/publications/statistics/koutsuu/opendata/index_opendata.html (accessed on 21 March 2022).

**Acknowledgments:** We thank Yuki Ishisaka for helping with the data analysis.

**Conflicts of Interest:** The authors declare no conflict of interest.

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
