# Peer review of "Characteristics of Cyclist Crashes Using Polytomous Latent Class Analysis and Bias-Reduced Logistic Regression"

_sustainability, doi:10.3390/su14095497_

Round 1

Reviewer 1 Report

The paper analyzed and classified cyclist crashes occurring in Japan between 2019 and 2020, then employed a latent class analysis using categorical indicators related to infrastructure, road users, vehicles, and environmental/time characteristics to classify cyclist crashes. The results showed that the variable contributing the most to the classification was the crash location. The topic is interesting, the research methodology is very practical and logical, and some of the findings obtained have significant contributions. However, there are a few problems to be further improved as well:

1) The literature review of cyclist crashes is not very adequate. The author needs to further enrich the literature review related to characteristics such as cyclist psychology.

2) Some parts of the paper need further evidence to make the content more convincing, for example:

In lines 311-321, the authors compare the categories of cyclist crashes in Japan, Denmark, and Italy. The standard rules of the road in Denmark and Italy are on the right, while in Japan it is on the left. The authors can try to analyze and discuss whether the rules will have an impact on the factor of cyclist crashes.

3) There are some grammatical and editorial issues in this paper. Please check and correct care.

Based on the topic, methodology and conclusions, I recommend that this paper be accepted with minor changes.

Author Response

Response to Reviewer 1 Comments

Point 1: The literature review of cyclist crashes is not very adequate. The author needs to further enrich the literature review related to characteristics such as cyclist psychology.

Response 1: Thanks for your kind reminder. We added four literatures related to cyclist psychology and re-numbered the references. We also added the following comments as one of the future tasks.

“Billot-Grasset et al. [15] pointed out that road user behavior influences each step in the chain of events leading to an accident. Ma et al. [16] showed that cyclists' crash risks were directly predicted by risky cycling behaviors and cycling anger and that the effects of cycling anger, impulsiveness, and normlessness on crash risks were mediated by cycling behaviors. These psychological aspects of cyclists and drivers are important for understanding accidents.”

 [Page 12, Line 444-9}

Point 2: In lines 311-321, the authors compare the categories of cyclist crashes in Japan, Denmark, and Italy. The standard rules of the road in Denmark and Italy are on the right, while in Japan it is on the left. The authors can try to analyze and discuss whether the rules will have an impact on the factor of cyclist crashes.

Response 2: Thanks for your kind reminder. We revised the sentences as follows.

“In Denmark and Italy, the driving rule are on the right, while in Japan it is on the left. However, this rule had no impact on classification.” [Page 9, Line 331-3}

Point 3: There are some grammatical and editorial issues in this paper. Please check and correct care.

Response 3: We went through the entire manuscript to eliminate grammatical mistakes.

Reviewer 2 Report

Interesting study about cyclist crashes using data from Japan. I have some comments for the Authors’ consideration as follows:

  1. Abstract section: it is beneficial for readers to state why this study is important, especially for international readers who are not familiar with cycling behavior in Japan.
  2. Section 2: please explain the reason for using only two years of the report (2019-2020) as the source of data. Please explain how these data were gathered and recorded. This is important for international readers to understand how this data was collected. It is also important to understand the meaning of the term used in recording/classification.
  3. Section 2: as many readers are not familiar with the system in Japan, please explain in more detail how the cyclist system in Japan. For example, I cannot understand the parameter to classify the severity. How serious is the fatality? What do you mean by urban DID? How dense it is? Etc.
  4. Line 139-140: it was stated that “….only 90,698 combinations are observed…” Please explained how the number of combinations is obtained.
  5. Section 3.1.2; please clarify which one is the black category and the blue category.
  6. Figure 2 and section 3.1; please consider to re-write the explanation of this figure. It is difficult to read the figure while reading the explanation is still confusing. For example, line 193-194, where it is difficult to notice the descending order of the fatality rate on the horizontal axis. Another example is lines 200-202, where I cannot find in the figure the percentage of observations.
  7. Section 4.1; what is the meaning of a different number of classes? Is it better? Similarly, what are the impact and the meaning of different contributing factors?
  8. Section 4.2 and 4.3; Please explain what is the lesson learned from this study for other locations or countries? What is your suggestion for geometric design?
  9. Conclusion; this study seems as having focused on Japan, while the explanation is not enough for readers to understand cycling behavior as well as geometric design for cyclists in Japan. Thus, what is the benefit for international readers after reading this study? As a piece of information is fine, but as an academic reference, what is the benefit. Please consider providing more discussion on practical implications, especially in designing cyclist facilities.

Author Response

Response to Reviewer 2 Comments

Point 1: Abstract section: it is beneficial for readers to state why this study is important, especially for international readers who are not familiar with cycling behavior in Japan.

Response 1: Thanks for your kind reminder. We added the following sentence and revised the abstract.

Although the number of cyclist crashes is decreasing in Japan, the fatality rate is not. Thus, reducing their severity is a major challenge. We used polytomous latent class analysis to understand their characteristics and bias-reduced logistic regression to analyze their severity.”

Point 2: Section 2: please explain the reason for using only two years of the report (2019-2020) as the source of data. Please explain how these data were gathered and recorded. This is important for international readers to understand how this data was collected. It is also important to understand the meaning of the term used in recording/classification.

Response 2: Thank you very much for the reminder. We added the following sentences.

“This database records accidents reported to the NPA. It is also all data publicly available in Japan, and none before 2019.” [Page 3, Line 128-9}

Point 3: Section 2: as many readers are not familiar with the system in Japan, please explain in more detail how the cyclist system in Japan. For example, I cannot understand the parameter to classify the severity. How serious is the fatality? What do you mean by urban DID? How dense it is? Etc.

Response 3: Thanks for your question. There are two levels of severity: crashes with and without fatalities. DID is the Densely Inhabited Districts which are a series of census districts with a population density of 4,000 or more per square kilometer and a population of more than 5000. We added these explanations in Section 2. [Page 3-4, Line 144-5 and Line 149-52}

Point 4: Line 139-140: it was stated that “….only 90,698 combinations are observed…” Please explained how the number of combinations is obtained.

Response 4: Thanks for your question. Each crash has 15 categorical variables. Summarising the accidents with the same attribute combinations, total number of combinations was 90,698.

Point 5: Section 3.1.2; please clarify which one is the black category and the blue category.

Response 5: Thank you very much for pointing this out. As the color is not important in this paper, we omitted this sentence. [Page 5, Line 194]

Point 6: Figure 2 and section 3.1; please consider to re-write the explanation of this figure. It is difficult to read the figure while reading the explanation is still confusing. For example, line 193-194, where it is difficult to notice the descending order of the fatality rate on the horizontal axis. Another example is lines 200-202, where I cannot find in the figure the percentage of observations.

Response 6: Thank you for your nice reminder. We revised the sentences in the followings.

“For each of these three locations, we re-numbered the classes in descending order of the fatality accident rates in each class. Vertical items were classified according to the road structure, environment, driver, and cyclist status. The categories for which significant differences were found for each variable are shown. If categories were spread across multiple classes, they were bundled together.” [Page 6, Line 199-203]

Point 7: Section 4.1; what is the meaning of a different number of classes? Is it better? Similarly, what are the impact and the meaning of different contributing factors?

Response 7: Thanks for your question. The number of classes is selected using BIC. [Page,9, Line 325] We can‘t compare the number of classes among countries, and discuss the impact and the meaning of different contributing factors because number of samples and variables for the poLCA are different among countries.

Point 8: Section 4.2 and 4.3; Please explain what is the lesson learned from this study for other locations or countries? What is your suggestion for geometric design?

Response 8: Thank you for your nice reminder. We added Section 4.2 for the lesson learned from this study.

“The installation of stop signs and median separations may reduce the risk of fatalities and should be promoted in the same manner as in other countries. However, the existence of a guard rail is another factor increasing the fatality risk.” [Page 10, Line 345-7]

We also revised Section 4.3 for the lesson learned from this study.

“The fatality risk on sidewalks is higher than that on prefectural roads. Of the 319 accidents, 14 were fatal, indicating a high fatality risk. The reason for this may be that the corresponding surfaces are less well-maintained than those of national and municipal roads. Because one of the parts of a bicycle involved in collisions is the front, it is conceivable that the holes in such sidewalks could lead to serious accidents. Uneven roads may exist as a result of insufficient road maintenance. Therefore, promoting these improvements may reduce the risk of death.” [Page 10, Line 389-95]

Point 9: Conclusion; this study seems as having focused on Japan, while the explanation is not enough for readers to understand cycling behavior as well as geometric design for cyclists in Japan. Thus, what is the benefit for international readers after reading this study? As a piece of information is fine, but as an academic reference, what is the benefit. Please consider providing more discussion on practical implications, especially in designing cyclist facilities.

Response 9: Thank you for your nice reminder. We added the following sentences.

“The results of this study suggest that it is important to implement countermeasures based on the crash location. At intersections, the installation of stop signs and median separation may reduce the risk of fatalities. It is common for cyclists to decelerate when approaching a stop sign and check to ensure that there is no vehicular or pedestrian cross traffic before proceeding through the intersection rather than putting a foot down. Many different parties can be liable for a bike accident at a stop sign, including the cyclist.

This study showed that median separation was one of the factors reducing severity, and the installation of guardrails potentially increases the risk of death. Wang et al. showed that tree separation for non-motorists increased the injury severity of e-bicyclists [35]. It would be worthwhile to investigate other separation treatments that could better protect e-bicyclists from crashes.

The presence of bumps and obstacles, particularly on rural roads with low pedestrian traffic, has rarely been reported to road managers, which may lead to an increase in single-vehicle accidents. Regular road maintenance of the shoulders and other non-roadway sections is important.” [Page 11-12, Line 415-29]

Reviewer 3 Report

The authors present a comprehensive analysis of ~140k road accidents involving cyclists. Using polytomuous latent class analysis 17 classes with distinct characteristics are found and described. The article is written concisely, provides a decent literature review section and the conclusions are sound, as far as I can tell. I'm not an expert in the field (most likely there was a mix-up as I work on Life-Cycle Analysis and not on Latent Class Analysis) but I can provide the outsiders view on the presented result. With this caveat, please consider the following comments.

- the authors write, p.2 l. 58f, "The class membership of each cyclist crash can then be inferred from the observed variables." From my limited understanding, I inferred that only probabilities can be stated for the class membership of the events.

- the authors write, p.4 l. 156 ff, "To ensure that the global maximum likelihood of the latent class model was achieved (rather than local maximum log-likelihood), we re-estimated each model 5,000 times and set 10 initial values of the class-conditional response probabilities; then, we saved the model with the greatest likelihood." How are these numbers chosen? Is there any indication that these numbers are sufficient to find a global maximum with a high confidence?

- a discussion of the weaknesses of the applied methods, in particular of the polytomuous latent class method, are missing.

Author Response

Response to Reviewer 3 Comments

Point 1: p.2 l. 58f, "The class membership of each cyclist crash can then be inferred from the observed variables." From my limited understanding, I inferred that only probabilities can be stated for the class membership of the events.

Response 1: Thanks for your comments. The classfication is based on the posterior probability that cases belong to each latent class. http://www2.uaem.mx/r-mirror/web/packages/poLCA/poLCA.pdf

Point 2: p.4 l. 156 ff, "To ensure that the global maximum likelihood of the latent class model was achieved (rather than local maximum log-likelihood), we re-estimated each model 5,000 times and set 10 initial values of the class-conditional response probabilities; then, we saved the model with the greatest likelihood." How are these numbers chosen? Is there any indication that these numbers are sufficient to find a global maximum with a high confidence?

Response 2: Thanks for your question. <R> package poLCA uses EM and Newton-Raphson algorithms to maximize the latent class model log-likelihood function. Depending on the starting parameters, this algorithm may only locate a local, rather than global, maximum. This becomes more and more of a problem as the number of classes increases. http://www2.uaem.mx/r-mirror/web/packages/poLCA/poLCA.pdf Therefore, we set 10 times to conduct the search for the global maximum. However, we do not know the sufficient numbers and their indicators.

Point 3: a discussion of the weaknesses of the applied methods, in particular of the polytomuous latent class method, are missing.

Response 3: Thanks for your kind reminder. We added the following sentences in conclusion.  

“Latent class analysis gives the best fit of a model to the data and do not give an indication of how well clusters are separated. The popularity of such methods might be explained by the apparent implication that results show clear (i.e. separate) clusters which are then assumed to represent a qualitatively distinct group of people with an existing disease entity. However, one should keep in mind that the classes formed are relative to each other and do not necessarily hold inherent meaning.” [Page.12, Line 430-6]

Reviewer 4 Report

Thanks for giving me the opportunity to review this article.

This is an interesting, well-written article of interest to the journal's audience.

Author Response

Response to Reviewer 4 Comments

Point 1: This is an interesting, well-written article of interest to the journal's audience.

Response 1: Thank you so much.

Round 2

Reviewer 2 Report

I am happy to read the revised version and thank you for the Authors’ effort in revising. I have a small comment for consideration to add clarity.

In the conclusion section, please discuss the limitation of this study which has an implication for the interpretation of the result. It is useful for international readers to notice when they conduct further research on this topic.

Author Response

Point 1: In the conclusion section, please discuss the limitation of this study which has an implication for the interpretation of the result. It is useful for international readers to notice when they conduct further research on this topic.

Response 1: Thanks for your comments. We revised the following sentences in conclusion adding three references.

“However, this study has the following issues regarding data and methods. The first challenge regarding the data limitations is the definition of severity. In this study, only two classifications were used: fatal or non-fatal. Fatalities are extremely rare, and future analysis by subdividing the injuries such as hospitalization or not may yield different results. In addition, the data collected in this study do not include detailed attributes of cyclists. Previous studies have analyzed factors such as alcohol consumption, helmet use and improper behaviors such as violation of the law [4, 5, 11]. It is important to take these factors into account in the analysis of fatality risk. Furthermore, the lack of traffic flow data for motor vehicles, cyclists, and pedestrians is a major deficiency. For example, the installation or removal of guardrails and utility poles should be considered not only for cyclists, but also for pedestrians and motor vehicles. In Japan, there are many cases of cyclists running on the wrong side of sidewalks and roadways, and automobiles parking on bicycle lanes. In urban road without bike lane, vehicle speeds with and without cyclists were found to be negligible [36]. And on-road bicycle lanes and parked cars reduced passing distance [37]. It is desirable to analyze these infrastructures on the severity combined with the mean and variances traffic volume by various road users and land use along roadsides data.

Regarding the methodology, latent class analysis provides the best fit when modeling the data, but does not provide an indication of how well clusters are separated. The popularity of such methods might be explained by the apparent implication that the results show clear (i.e., separate) clusters, which are then assumed to represent a qualitatively distinct group of people with an existing disease entity. However, it should be noted that the classes formed are relative to each other and do not necessarily have an inherent meaning.

Finally, Billot-Grasset et al. [15] pointed out that road user behavior influences each step in the chain of events leading to an accident. Ma et al. [16] showed that cyclists' crash risks were directly predicted by risky cycling behaviors and cycling anger and that the effects of cycling anger, impulsiveness, and normlessness on crash risks were mediated by cycling behaviors. These psychological aspects of cyclists and drivers are important for understanding accidents. The development of bicycle space will ultimately lead to the creation of a safe travel space for all road users [38]. Although under-reporting biases have been noted [39], not simply the severity but also the probability of accidents also needs to be examined further.” [Page.11, Line 430-62]